# The Probability of Ischaemic Stroke Prediction with a Multi-Neural-Network Model

**DOI:** 10.3390/s20174995

**Published:** 2020-09-03

**Authors:** Yan Liu, Bo Yin, Yanping Cong

**Affiliations:** 1College of information science and engineering, Ocean University of China, Qingdao 266000, China; yanliukd@163.com (Y.L.); congyp@ouc.edu.cn (Y.C.); 2Pilot National Laboratory for Marine Science and Technology, Qingdao 266000, China

**Keywords:** convolutional neural network, stroke, feature extraction, disease diagnosis

## Abstract

As is known, cerebral stroke has become one of the main diseases endangering people’s health; ischaemic strokes accounts for approximately 85% of cerebral strokes. According to research, early prediction and prevention can effectively reduce the incidence rate of the disease. However, it is difficult to predict the ischaemic stroke because the data related to the disease are multi-modal. To achieve high accuracy of prediction and combine the stroke risk predictors obtained by previous researchers, a method for predicting the probability of stroke occurrence based on a multi-model fusion convolutional neural network structure is proposed. In such a way, the accuracy of ischaemic stroke prediction is improved by processing multi-modal data through multiple end-to-end neural networks. In this method, the feature extraction of structured data (age, gender, history of hypertension, etc.) and streaming data (heart rate, blood pressure, etc.) based on a convolutional neural network is first realized. A neural network model for feature fusion is then constructed to realize the feature fusion of structured data and streaming data. Finally, a predictive model for predicting the probability of stroke is obtained by training. As shown in the experimental results, the accuracy of ischaemic stroke prediction reached 98.53%. Such a high prediction accuracy will be helpful for preventing the occurrence of stroke.

## 1. Introduction

### 1.1. Related Work

Stroke is one of the main causes of death and disability worldwide [1]. Due to lack of effective treatment, it is difficult to cure stroke patients completely. On the other hand, even if the patient is cured, they still have to face harsh realities: permanent disability, incapacity [2], reduced social activities [3], etc. Hence, the disease puts a heavy burden on patients, the health care system and society. According to lots of studies, there is a certain eclipse period before the onset of stroke and early prediction and prevention can effectively reduce incidence rate of the disease. Actually, some premonitory symptoms appear during the eclipse period of stroke. For example, Zhang designed a questionnaire to put forward that a series of symptoms will appear during the eclipse period before the onset of stroke [4], such as chronic yawning, frequent choking coughs and a habit of biting the tongue. Goldstein [5] has divided risk factors into nonmodifiable risk factors (age, gender, race and genetic, etc.), well-documented and modifiable risk factors (high blood pressure, smoking, diabetes, atrial fibrillation, some other heart diseases, etc.) and less well-documented or potentially modifiable risk factors (metabolic syndrome, alcoholism, drug abuse, etc.), and it was suggested that changing well-documented and modifiable risk factors or less well-documented or potentially modifiable risk factors could reduce the risk of stroke onset. Therefore, early detection and prevention can effectively reduce the risk of ischaemic stroke and increase the success rate of cure during the eclipse period.

In the medical field, a large-scale independent electronic health record (EHR) database has been established, which provides a large quantity of clinical diagnoses, and imaging and laboratory data [6]. It makes a considerable contribution to predicting the occurrence of diseases by artificial intelligence, which has been widely used for analysis and prediction, with remarkable results. In this field, it is used to deal with complex disease prediction tasks [7]. For example, Czabanski used the Lagrangian support vector machine to predict atrial fibrillation (AF). The results obtained during the test stage showed that the classification accuracy was 98.86%; it can effectively detect AF and provide more reliable information for the processing stage after the onset of AF [8]. Osman used an automatic epilepsy diagnostic method based on a self-organization map (SOM) method to discover epilepsy [9], and the detection accuracy of the model reached 97.47%; it could effectively detect epilepsy. Therefore, the combination of deep learning and big data has made remarkable achievements in the field of disease prediction.

In terms of the prediction of stroke, many researchers used artificial intelligence technology to predict stroke. For example, Songhee [10] used a deep neural network based on extended PCA to extract features from medical service usage and health behavior data and predicted stroke; the area under the curve (AUC) value of our method was 83.48%. It can be used by both patients and doctors to prescreen for possible strokes; however, the risk factors considered are not comprehensive, and the predictive performance of the model is average. Chen-Ying [11] used a deep learning network model to perform feature extraction and stroke prediction on electronic medical claim records. The area under the curve (AUC) value of the method was 0.915. The prediction performance of the prediction model is good. However, there is still a problem of insufficient consideration of factors related to stroke. Although the above-mentioned researchers proved that using deep learning techniques can predict stroke, they all used single-modal data for model training and prediction, and the accuracy of the model is average.

In addition, many researchers have explored many different predictors related to stroke, which provide a sufficient feasibility basis for the prediction methods of stroke. Flint, a stroke specialist at Kaiser Permanente medical center in the United States, performed 36 million blood pressure measurements on more than one million people. This study brought a large amount of data and provided a clear answer to the basic question about blood pressure and stroke: "diastolic blood pressure" and "systolic blood pressure" are independent predictors of stroke risk [12]. Wesley collected left ventricular hypertrophy (LVH) detected by electrocardiography (ECG-LVH) and LVH detected by echocardiography to assess the risk of stroke. Finally, ECG and echocardiography were found to be predictive factors for stroke [13]. Di [14] recruited 11 post-stroke patients and 20 healthy control subjects and performed an elbow sinusoidal trajectory tracking experiment. The experimental results showed that the EMG’s fApEn (fuzzy approximate entropy) values of the experimental group and the control group were significantly different, so stroke can induce neurological changes in paretic muscles. Bodapati examined that whether 24-hour heart rate variability (HRV) added predictive value to the Cardiovascular Health Study clinical stroke risk score (CHS-SCORE) [15]. The value of adding HRV to the CHS-SCORE was assessed with stepwise Cox regression analysis. They found that two HRV parameters, CV% (coefficient of variance of NN intervals) and power law slope, emerged as significantly associated with incident stroke when added to a validated clinical risk score. Chantamit-o-pas et al. integrated the icd-10 code into the health records and other potential risk factors in Electronic Healthcare Records (EHRs) into the patterns and models to predict stroke [16].

### 1.2. Novelty and Contributions

Current forecasting methods mainly use single-modal data, and the field of stroke prediction is no exception. If the features in text, image or stream data are needed, some methods are used to extract the required features from the unstructured data. How to process multi-modal data with the help of multiple end-to-end neural networks, and to make fusions and predictions, are very important technical challenges.

To overcome those challenges, we combined the stroke risk predictors obtained by previous researchers and propose a multi-model fusion convolutional neural network architecture to predict the occurrence probability of ischemic stroke (here the stroke referred to is ischemic stroke). In this method, a convolutional neural network is the first part used for feature extraction. In fact, there are two models here. One of them is convolutional neural network based on the VGG16 model used to extract features from an electrocardiogram (ECG), an electromyogram (EMG), a blood pressure graph and a heart rate graph. The other is a one-dimensional convolutional neural network model, which is used to extract features from personal health information (smoking, drinking, history of atrial fibrillation, history of hyperlipidemia, etc). The second part fuses all the features acquired in the first part and makes stroke predictions. This study used multiple end-to-end models to fuse and predict multi-modal data of all stroke-related predictors; we also solved the problem of multi-modal data fusion in disease prediction.

## 2. Materials and Method

### 2.1. Dataset

Since stroke is accompanied by dynamic cerebral automatic regulation injury, the factors related to the state of dynamic cerebral automatic regulation injury, such as blood pressure, heart rate, ECG and EMG, can be used to predict stroke. The public dataset called cerebral vasoregulation in elderly with stroke [17], published by Goldberger [18], is used here. It contains data from a large multi-modal study that investigated the effects of ischaemic stroke on cerebral vascular regulation. The cross-sectional study compared 60 subjects who suffered strokes to 60 control subjects, collecting the following data for each patient and normal person across multiple days: transcranial doppler of cerebral arteries; 24-hour blood pressure numerics; high resolution waveforms (ECG, blood pressure, CO2 and respiration) during various movement tasks; 24-hour ECG, EMG, and accelerometer recordings; and gait pressure recordings during a walking test. The parts of the human body detected by ME6000 are shown in Table 1. As the information of some research subjects is incomplete, the data from 39 patients with ischaemic stroke and 40 normal persons were chosen. Demographic characteristics among the two groups are shown in Table 2. Blood pressure, heart rate, ECG and EMG and personal health information (smoking, drinking, history of atrial fibrillation, history of hyperlipidaemia, etc.) are used in this paper; baseline information about stroke patients and normal persons is shown in Table A1, Table A2, Table A3,Table A4,Table A5, Table A6, Table A7, Table A8, Table A9, Table A10, Table A11 and Table A12.

These factors can reflect the impacts of stroke on cardiovascular and cerebrovascular diseases. Hence, the current state of cardiovascular and cerebrovascular diseases can be judged by observing the factors. On the other hand, since personal health data such as smoking [19], alcoholism [20], history of hypertension [21] and history of hyperlipidaemia have strong correlations with stroke [22], it is necessary to combine personal health data to infer the possibility and probability of stroke attack. Table 3 lists the predictors related to stroke used in this article.

The data used in the experiment can be divided into two types, streaming data and structured data. The streaming data contain blood pressure, heart rate, ECG and EMG data. Twenty-four-hour beat-to-beat heart rate and BP monitoring using Dynapulse were measured at 20 min intervals during daytime and at 30 min intervals at night; 24-hour ECG and EMG monitoring was done using ME6000 devices. ECG and EMG data were sampled at 1000 Hz. ECG and EMG were done for 24 h during sleep and daily activities, such as walking; therefore, both ECG and EMG data have two types, static state data and dynamic data. The structured data are the personal health information (age, gender, height, history of hypertension, alcoholism, smoking, etc.) of 79 subjects.

For streaming data, graphical and integrated processing was carried out. First, streaming data were converted into graphs, such as a blood pressure graph (as shown in Figure A2), a heart rate graph (as shown in Figure A3), an ECG and an EMG (as shown in Figure A1). In order to easily show the characteristics of all streaming data, the blood pressure graph (upper left), heart rate graph (upper right), an ECG and an EMG (lower left and right) are shown in Figure 1.

One-dimensional data were obtained by transforming and processing the structured data. Some of the information (gender, history of hypertension, family members with histories of hypertension, etc.) has two or more values, so it needs to be processed by one-hot encoding [23]. For example, in Table 4, index values from 7 to 9 indicate race types (African American, White, and Asian). The details are shown in Table 4 and Table 5, which respectively represent data before and after personal health conversion.

### 2.2. Experimental Design

#### 2.2.1. Overall Architecture of Proposed Model

The network model proposed in this paper mainly includes two parts. In the first part, a convolutional neural network model based on VGG16 was built to extract the features of blood pressure, heart rate, EMG and ECG, because the VGG16 model has a better classification effect in image classification [24,25]. For feature extraction of personal health data, the one-dimensional conolutional neural network model was built. In the second part, a model for feature fusion was built to train the prediction model that can predict the occurrence probability of stroke. The entire network structure system is shown in Figure 2.

#### 2.2.2. The First Part: Feature Extraction

The first model used a model based on convolutional neural networks to distinguish the waveforms of sign-of-life parameters of stroke patients. The model includes a VGG16 model; a fully connected layer with 256 neurons and the ReLU, which is used as the activation function; and a fully connected layer with a neuron and the softmax, which is used as the activation function. The work flow of this model is as follows: First, input the waveform graphs with the shape of 150×150×3 into the model to train the model. Features of the waveform graphs are extracted through the VGG16 model, and then input into the fully connected layer of the next layer to get a feature layer with a shape of 256×1 for next step.

The second model was built using a one-dimensional convolutional neural network model [26] to identify the personal health data of stroke patients. The model consists of two layers of one-dimensional convolutional layers containing 16 and 32 neurons, and ReLU is used as the activation function; two pooling layers—one fully connected layer containing a neuron, and sigmoid is used as the activation function. The work flow of this model is as follows: First, the convolutional layer and the pooling layer are used to convolve and pool the text to extract features, and then the features are input into the fully connection layer of the next layer, and the feature layer with a shape of 32×1 is obtained for the next step.

#### 2.2.3. The Second Part: Prediction of Incidence Probability of Stroke

The third model was composed of four layers of fully connected layer. The numbers of neurons in the first three layers of the fully connected layer were 64, 32 and 16. ReLU is used as the activation function. The number of neurons in the last layer was 1, and the activation function was softmax. The two models obtained from the training in first part were used to obtain feature layers with shapes of 256×1 and 32×1. Then the two feature layers were fused [27] to obtain the 189,679 one-dimensional dataset with a shape of 288×1, which was then used to train the model for predicting the incidence probability of stroke.

## 3. Results

During the experiment, graphs of sign-of-life parameters and 79 one-dimensional data with the shape of 70×1 were used to train the model for extracting the feature from stream data and the model for extracting the feature from structured data in the first part. In the second part, the two feature layers were combined as the training data input layer of the model for feature fusion. Finally, a prediction model was obtained to predict the probability of stroke.

### 3.1. Results of Training a Model for Extracting Features from Streaming Data

Graphs of sign-of-life parameters were divided into three sets, a training set (60%), a verification set (20%) and a testing set (20%). The training set and verification set were used to train the model for extracting features from streaming data. The optimizer was RMSProp, the learning rate was 1 × 10^−5^, the loss function was binary cross entropy and the number of iterations of the training was 30. Finally, the model for extracting features from streaming data was able to identify the waveform graphs of sign-of-life parameters of stroke patients and normal persons. The curves of accuracy and loss rate of the feature extraction model are shown in Figure 3 and Figure 4.

### 3.2. Results of Training a Model for Extracting Features from Structured Data

The one-dimensional dataset was divided into three sets, a training set (60%), a verification set (20%) and a testing set (20%). The training set and verification set were used to train the model for extracting features from structured data. The optimizer was RMSProp, the learning rate was 1 × 10^−4^, the loss function was binary cross entropy and the number of iterations of the training was 100. Finally, a classified prediction model that could identify personal health data of stroke patients and normal persons was obtained. The curves of accuracy and loss rate of the feature extraction model are shown in Figure 5 and Figure 6.

### 3.3. Results of Training a Model for Feature Fusion

On the one hand, graphs of signs of life were input into the feature extraction model for streaming data; a 256×1 feature dataset was then obtained. On the other hand, the one-hot encoded personal health data of 79 subjects were input into the feature extraction model for structured data to obtain the 32×1 feature data. Then the feature layer of the sign-of-life parameters of the research subjects was combined with the corresponding feature layer of the personal health data to obtain a set of feature layers with a shape of 288×1, which were used as the training data of the model for feature fusion.

The combined feature layer set obtained above was divided into three categories: a training set (60%), a verification set (20%) and a testing set (20%). The training set and verification set were used to train the feature fusion model. The optimizer was Adam and the loss function was categorical cross entropy. The number of training iterations was 3000. Finally, a prediction model for feature fusion capable of predicting the incidence probability of stroke was obtained. The curves of the accuracy and loss rate of the feature fusion model are shown in Figure 7 and Figure 8.

### 3.4. Model Evaluation

The testing group containing 38,808 samples was used to evaluate the model for feature fusion, and the confusion matrix obtained is shown in Figure 9. Due to the value of AUC being up to 0.99, the prediction performance of the proposed model was proven, as shown in Figure 10. The index values of the model evaluation (precision, recall, accuracy, AUC and f1-score) are shown in Table 6; the precision, recall and f1-score were obtained by Equations (1)–(3). As os known, the accuracy rate is expressed as the proportion of positive samples predicted as positive samples. There are two cases for predicting positive samples. One case labeled as TP is to predict positive samples as positive samples, and the other labeled as FP is to predict negative samples as positive samples. In this paper, a positive sample represents a stroke patient and a negative sample represents a normal person. Obviously, the precision of the proposed model was 98.59%. In the other hand, the accuracy of the proposed model, which is expressed as the proportion of samples that are correctly predicted, was up to 98.53%. In short, these results justify the fact that the proposed model has a good performance in terms of distinguishing the sign-of-life parameter waveforms and personal health information of stroke patients and normal people, and predicting the probability of a stroke patient, that is, the occurrence probability of stroke.

Once the predictive model recognizes that the sign-of-life parameters and personal health data of the current test subject have the characteristics of a stroke patient, the current test subject is judged to be a stroke patient. If the current test subject has not had a stroke, the result will be used as a pre-stroke warning. At this time, the subject should take corresponding preventive measures in time to reduce the harm caused by stroke.
(1)Precision=TP/(TP+FP)
(2)Recall=TP/(TP+FN)
(3)f1−score=(2×Precision×Recall)/(Precision+Recall)

## 4. Discussion

In the model for extracting features from streaming data, the convolutional neural network model based on VGG16 was used to extract the features of ECG, EMG, blood pressure and heart rate to identify the waveform graphs of sign-of-life parameters of stroke patients. It should be noted that VGG16 did have perform better when extracting features compared with other models, such as VGG19, DenseNet201 and ResNet50, as shown in Figure 11 and Table 7. According to the results, the recognition accuracy of VGG19 was the worst. Further, the training time of VGG16 was the shortest compared with DenseNet201 and ResNet50. Meanwhile, the number of parameters was the least. Due to these reasons, VGG16 was selected as the basic network model for extracting features from the streaming data.

The multiple end-to-end network models proposed in this paper realized the feature fusion of multi-modal data and stroke prediction. We compared the method proposed in this paper with the current stroke prediction methods [10,11], as shown in Table 8. First, the method proposed in this paper has made perfect measures in terms of input data, changing from universal single-modal data to multi-modal data. Secondly, optimization was made on the network model, and a prediction model based on multi-model fusion was used to extract and fuse multi-modal data. Finally, a stroke prediction model with better classification performance than other methods was obtained. This model is used to identify the abnormal characteristics of stroke in the sign-of-life parameters and personal health data in time, so as to prepare for stroke prevention measures in advance to reduce the harm caused by stroke.

## 5. Conclusions

The purpose of the current study was to estimate the probability of stroke occurrence. Hence, a convolutional neural network based on multi-model fusion was proposed. First, feature extraction of streaming data and structured data was carried out in combination with a convolutional neural network to expand ischaemic stroke-related factors and enhance feature extraction ability. Second, this paper proposed the processing of multi-modal data by multiple end-to-end neural architectures to achieve feature fusion and stroke prediction, and solved a major technical problem in disease prediction, which effectively improved upon the prediction accuracy of traditional models. To verify the effectiveness of the proposed method, the personal health data of 79 subjects were used in experiments that were carried out (based on a public dataset). The prediction accuracy reached 98.53%. This study contributes to our understanding of the impacts of risk factors on the occurrence of stroke. It could be used to help detect the disease early and thereby help institute appropriate control measures.

## Figures and Tables

**Figure 1 sensors-20-04995-f001:**
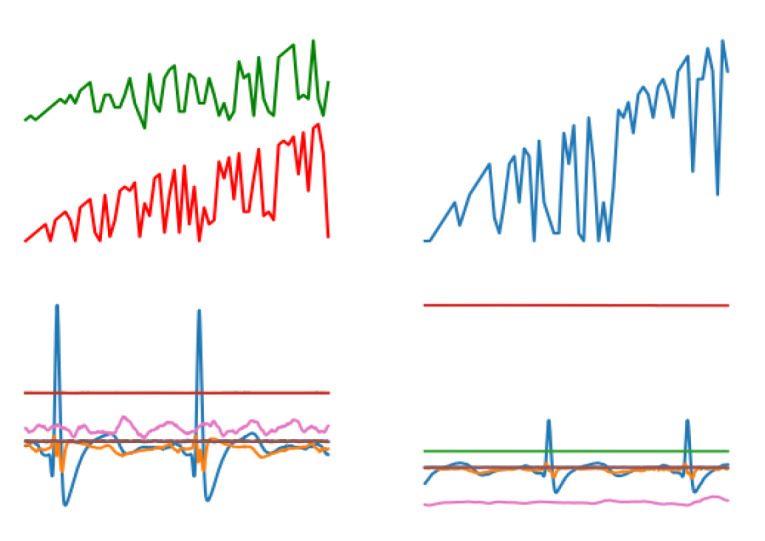
The composite image contains a blood pressure graph, a heart rate graph, an ECG and an EMG.

**Figure 2 sensors-20-04995-f002:**
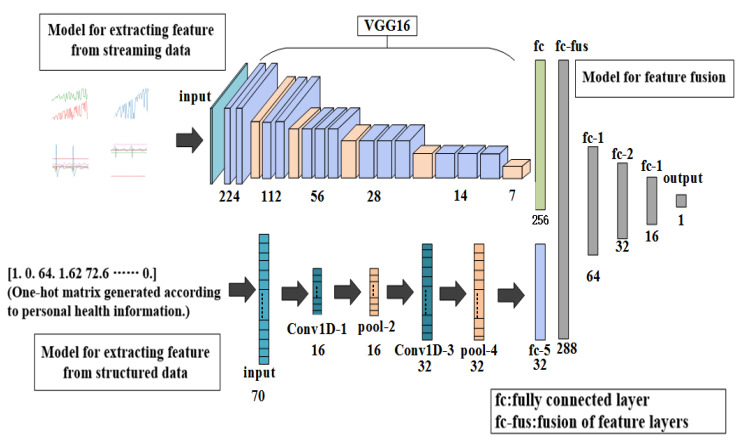
Overall architecture of proposed model.

**Figure 3 sensors-20-04995-f003:**
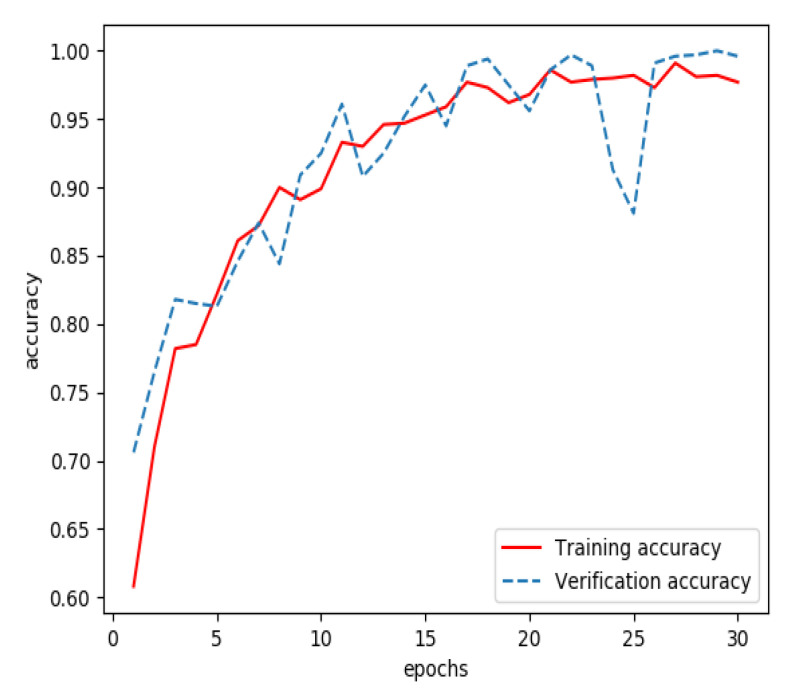
Accuracy of the model for extracting features from streaming data during training.

**Figure 4 sensors-20-04995-f004:**
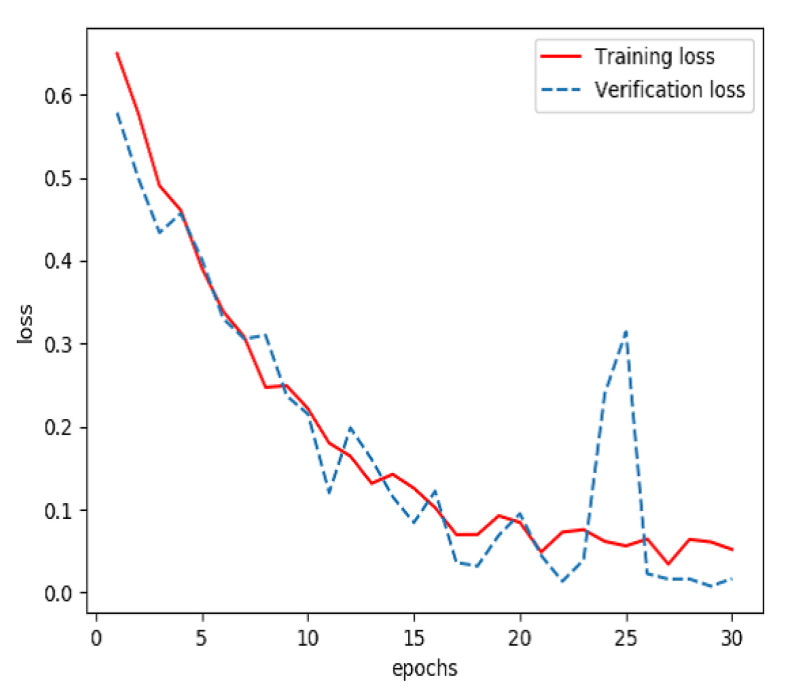
Loss of the model for extracting features from streaming data during training.

**Figure 5 sensors-20-04995-f005:**
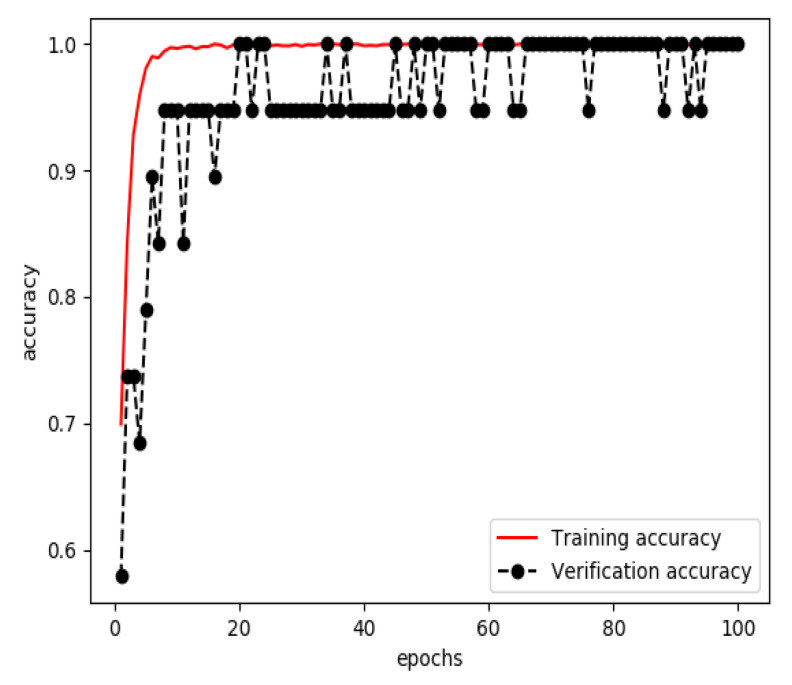
Accuracy of the model for extracting features from structured data during training.

**Figure 6 sensors-20-04995-f006:**
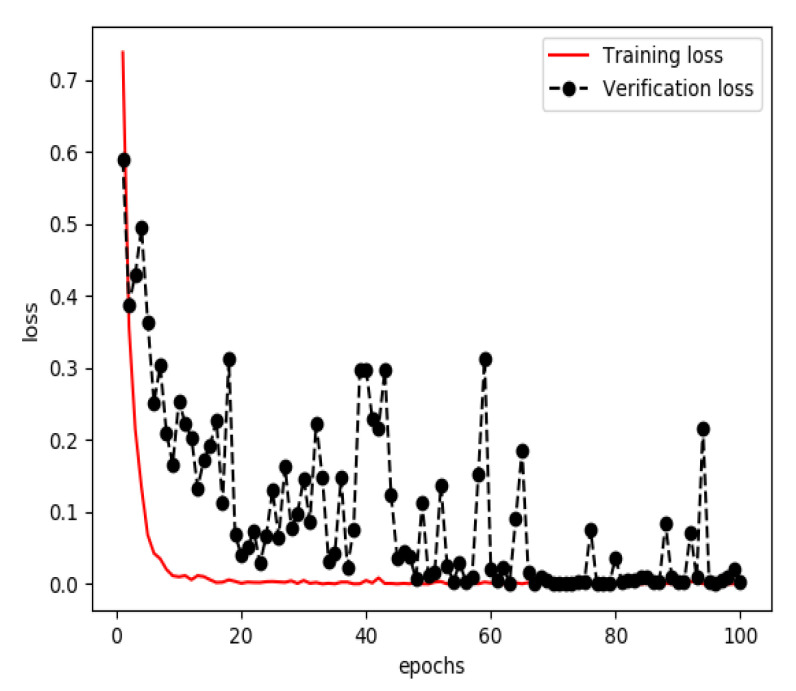
Loss of the model for extracting features from structured data during training.

**Figure 7 sensors-20-04995-f007:**
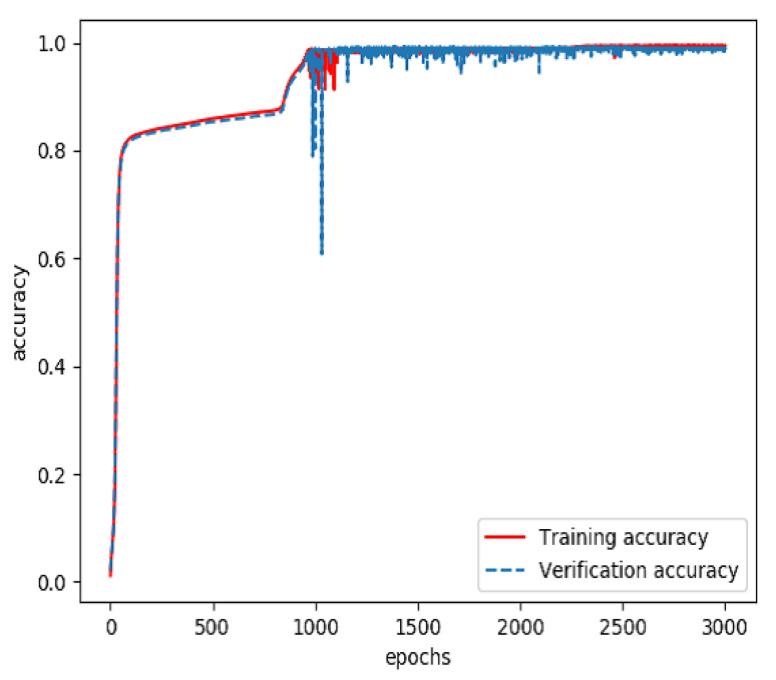
Accuracy of the model for feature fusion during training.

**Figure 8 sensors-20-04995-f008:**
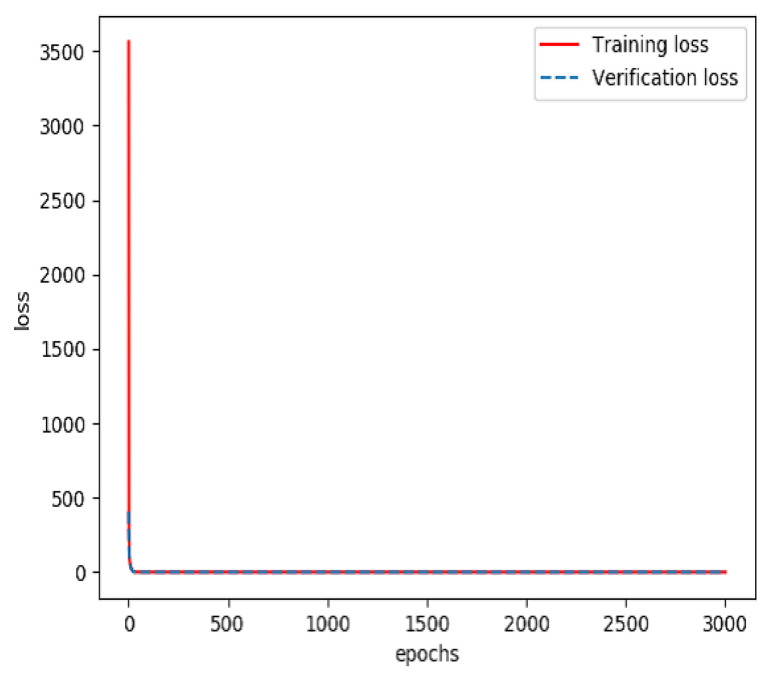
Loss rate of the model for feature fusion during training.

**Figure 9 sensors-20-04995-f009:**
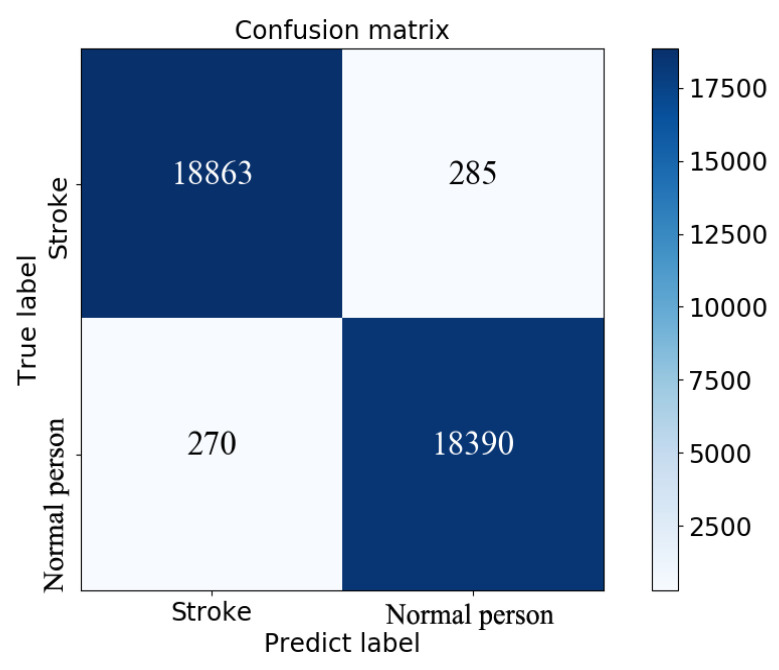
Confusion matrix obtained by model evaluation on the testing set.

**Figure 10 sensors-20-04995-f010:**
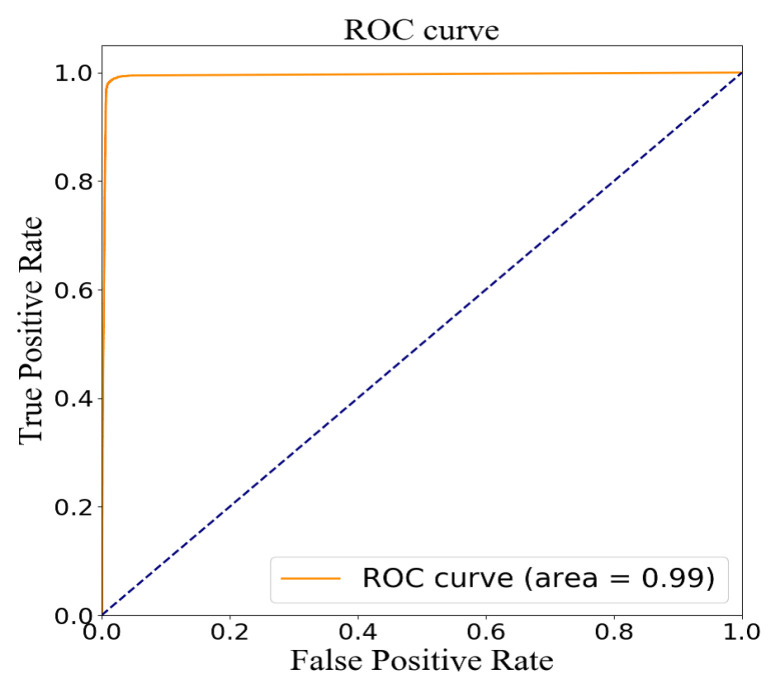
ROC curve by model evaluation on the testing set.

**Figure 11 sensors-20-04995-f011:**
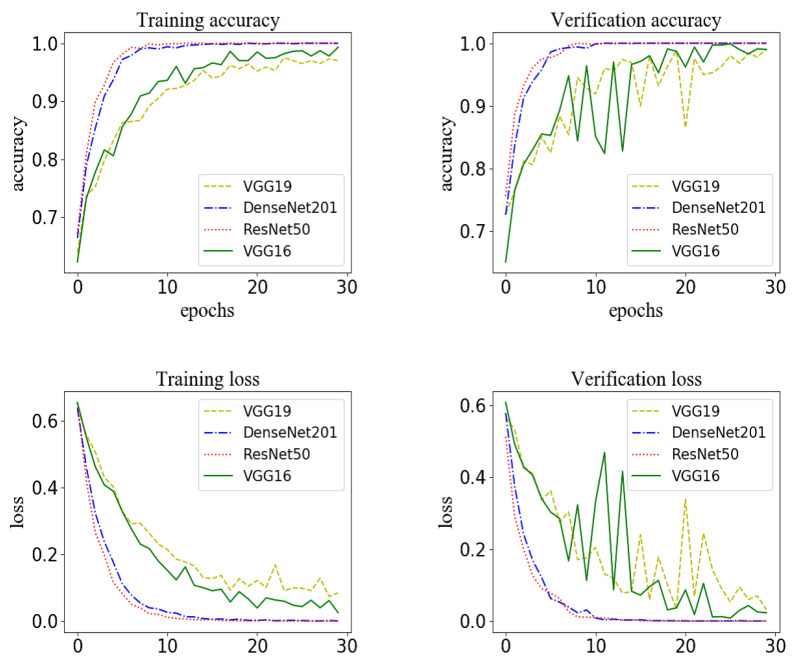
The model for extracting features from streaming data based on different network structures: the accuracy and loss for the training set and the accuracy and loss for the verification set.

**Table 1 sensors-20-04995-t001:** The parts of the human body detected by ME6000.

Type	Position
ECG 1	CH1 V5/V6-L clavicle
ECG 2	CH2 V1/V2 L clavicle
EMG 1	gastrocnemius right
EMG 2	gastrocnemius left

**Table 2 sensors-20-04995-t002:** Demographic characteristics among the two groups.

Group	Stroke	Control	*p*
Age (years)	64.21 (±8.94)	64.48 (±8.07)	0.87
Sex (male, female)	20, 19 (39)	17, 23 (40)	N = 79
Race (W, A, AA)	33, 1, 5	33, 3, 4	
Body mass index (kg/m^2^)	27.53 (±4.74)	27.59 (±6.48)	0.95
Years after stroke	6.05 (±4.88)	-	-
Stroke side (right, left)	24, 19	-	-
Infarct volume (cm^3^)	18.69 (±34.06)	-	-
NIHSS	2.71 (±2.72)	-	-
MRS	1.2 (±1.14)	-	-

Continuous variables are presented as mean ± SD, Ordinal variables are presented as mean ± SD (range), Nominal variables are presented as numbers, Comparison is not significantly different if *p* > 0.05, Race: W—White, A—Asian, AA—African American.

**Table 3 sensors-20-04995-t003:** Influencing factors of stroke.

Human Characteristic Data	Age Mass/kg Gender	Height/m BMI Race
Personal medical history	Htn patient medical history	Neuropathy autonomic symptoms
	Dizziness autonomic symptoms	Numbness autonomic symptoms
	DM/on-DM stroke	Syncope autonomic symptoms
	OHspecific autonomic symptoms	Atrial fibirillation patient medical history
	HTN years patient medical history	Cancer patient medical history
	Stroke patient medical history	DM patient medical history
	Heart failure = CHF /ifaction = -MI	
	patient medical history	
Behavioral	Current tobacco use	Pevious tobacco use
	Previous alcohol use	Pack tobacco years
	ALCOHOL Dose/Week	
Family medical history	Cancer family history	Cancerspecific family history
	HeartDisease family history	Hdspecific family history
	HTN family history	HTNspecific family history
	DM family history	Dmspecific family history
	Stroke family history	StrokeSpecific family history
Life sign parameters	Heart rate	blood pressure
	ECG	EMG

**Table 4 sensors-20-04995-t004:** One example of one-hot encoding of personal health data information.

Index	Value									
0–9	1.0	0.0	64.0	1.63	72.6	27.5	0.0	1.0	0.0	0.0
10–19	0.0	0.0	0.0	0.0	1.0	0.0	0.0	0.0	0.0	0.0
20–29	0.0	0.0	0.0	0.0	0.0	0.0	0.0	0.0	0.0	0.0
30–39	0.0	0.0	0.0	0.0	0.0	0.0	0.0	0.0	0.0	0.0
40–49	0.0	0.0	0.0	0.0	0.0	0.0	0.0	0.0	0.0	0.0
50–59	0.0	0.0	0.0	0.0	0.0	0.0	0.0	0.0	0.0	0.0
60–69	0.0	0.0	4.0	0.0	0.0	0.0	0.0	0.0	0.0	0.0

**Table 5 sensors-20-04995-t005:** The values of personal health data information before conversion.

Factor	Value	Used One-Hot
Htn patient medical history	YES	NO
Age	70	NO
Alcohol Dose/Week	0	NO
Neuropathy autonomic symptoms	YES	NO
Previous Tobacco Use	YES	NO
Current Tobacco Use	NO	NO
HeartDisease family history	1	NO
HdspeciFIc family history	f	YES
Stroke year patient medical history	0	NO
Atrial FIbtrillation patient medical history	NO	NO
BMI	26.7	NO
Gender	F	NO
Painful feet autonomic symptoms	NO	NO
Syncope autonomic symptoms	NO	NO
cancSpec family history	NULL	NO
HTN years patient medical history	4	NO
DM patient history	0	NO
DmspeciFIc patient history	NULL	YES
DM patient medical history	NO	NO
Height/m	1.64	NO
Mass/kg	71.67	NO
Dizziness autonomic symptoms	NO	NO
Numbness autonomic symptoms	NO	NO
Pack years	20	NO
Previous alcohol use	YES	NO
HTN family history	0	YES
HTNspeciFIc family history	NULL	YES
Heart failure = CHF/ifaction = -MI patient medical history	NO	NO
Race	White	YES
DM Non-DM stroke	Non-DM	NO
OH autonomic symptoms	NO	NO
Cancer family history	0	YES
Cancer patient medical history	NO	NO
Stroke patient medical history	NO	NO
Stroke family history	0	NO
Stroke Specific family history	NULL	YES

**Table 6 sensors-20-04995-t006:** Evaluation index values of the model.

TP	FN	FP	TN	Precision	Recall	Accuracy	AUC	*f*1-Score (0)	*f*1-Score (1)
18863	285	270	18390	98.59%	98.51%	98.53%	0.99	0.96	0.96

**Table 7 sensors-20-04995-t007:** Comparison of training time, number of parameters and accuracy using different network structures on testing sets.

	Accuracy	Training Time (Second)	Total Parameters
VGG19	0.96	1678	122122049
DenseNet201	0.97	34271	26186817
ResNet50	0.97	20162	23638913
VGG16	0.97	12689	16812353

**Table 8 sensors-20-04995-t008:** The method proposed in this paper with the current stroke prediction methods.

Methods	Input Data	Model Structure	AUC
DNN with scaled PCA	Medical service use and health behavior data	DNN	83.48%
Deep neural network	Electronic medical claims (EMCs)	DNN	91.5%
Multi model	Streaming data (Blood pressure etc.), structured data (EHRs)	Multi model fusion	99%

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
