# Peer review of "The Probability of Ischaemic Stroke Prediction with a Multi-Neural-Network Model"

_sensors, 2020, doi:10.3390/s20174995_

Round 1

Reviewer 1 Report

The authors present a study examining the detection of ischemic stroke using a neural network model fusion model that takes into account a variety of risk factors. This paper addresses an important need for individuals to assess stroke using two primary inputs, streaming data and patient history/constitutional data.

Unfortunately, this draft is missing much data that prevent it from being a meaningful contribution. If the authors took more time addressing details and characteristics of the patient population, details of the input, sparsity of input and focused less on evaluating the nuances of loss and accuracy with epochs, there could be value. However, as it stands now, the results, conclusions and experimental design are unclear, in addition to a non-traditional method of writing.

I would suggest the following: In the subset of 79 participants fully detail available data in a summary table, with full detail in supplemental information. Use the NN framework developed to specifically nominate an outcome measure with a clear endpoint (i.e. ischemic stroke within N months). Go into greater detail about the clinical questionare or information (what is the interval between streaming data and acquisition of clinical data), or even group clinical data into categories (cardiovascular, endocrine, behavioral etc...).

As it stands now this manuscript is difficult to follow and would merit from a full revision to focus on what needs to be examined, after which I would be glad to see the results of this fusion network be properly evaluated.

Author Response

Dear reviewer,

Many thanks for these valuable suggestions. According to your suggestions, we revised the manuscript. The changes we made are as follows:

#1. We have listed fully detail available data of 79 participants in Table A1 to Table A12 on pages 13 to 22.

#2. We listed the details and characteristics of 79 participants in Table 2 on page 4.

#3. In Table 3. We have grouped clinical data into four categories (Personal medical history, family medical history, human characteristic, behavioral) on page 5.

#4. The prediction results of the prediction model proposed in the article are introduced in detail, at the same time, the ‘Result’ has been revised on page 10, lines 221-225: Once the predictive model recognizes that the life sign parameters and personal health data of the current test subject have the characteristics of a stroke patient, the current test subject is judged to be a stroke patient. If the current test subject has not had a stroke, the result will be used as a pre-stroke warning. At this time, the subject should take corresponding preventive measures in time to reduce the harm caused by stroke.

In addition, we have revised the overall structure of the paper, and the revised content is the highlighted part of the latest version of the manuscript named " sensors-898789-r01.pdf".

Kind regards,

Yan Liu

Reviewer 2 Report

The paper applies deep learning to predict probability of ischemic strokes. The paper is written relatively well. English could be significantly improved. Some corrections (but not all) are indicated in the attached pdf file. The main problem is the lack of comparison with the state-of-the-art results and motivation of the proposed approach. Also, considering the results discussed in the literature (see Section 1.1), the accuracy indicated there (see [8,9]) exceeded the results presented by the Authors casting doubt on the value of the proposed approach. The proposed approach needs to be better motivated. 

Author Response

Dear reviewer,

Many thanks for these valuable suggestions. According to your suggestions, we revised the manuscript. The changes we made are as follows:

#1. We revised the formatting errors and some grammatical errors in the manuscript.

#2.  In the ‘Discussion’ section on pages 11 to 12,  lines 235 to 243, we put forward a comparison between the method proposed in this paper and the state-of-the-art methods, For convenience, the updated part is as follows:

The multiple end-to-end network models proposed in this paper realized feature fusion of multi-modal data and stroke prediction. Compared the method proposed in this paper with the current stroke prediction methods [10]-[11], as shown in Table 8. First, the method proposed in this paper has made perfect measures in terms of input data, changing from universal single-modal data to multi-modal data. Secondly, optimization was made on the network model, and a prediction model based on multi-model fusion was used to extract and fuse multi-modal data. Finally, a stroke prediction model with better classification performance than other methods was obtained. This model is used to identify the abnormal characteristics of stroke in the life sign parameters and personal health data in time, so as to prepare for stroke prevention measures in advance to reduce the harm caused by stroke.

#3. In addition, references 8 and 9 are used to predict atrial fibrillation and epilepsy respectively. The purpose of citing these two references in section 1.1 is to illustrate that deep learning has made a very fruitful contribution to the field of disease prediction. Therefore, these two citations can convince readers that using deep learning to predict stroke can also achieve similar accuracy.

  1. Czabanski, Horoba,Wrobel, Matonia, and Leski, J. Detection of Atrial Fibrillation Episodes in Long-Term Heart Rhythm Signals Using a Support Vector Machine. Sensors 2020, vol. 20, no. 3, p. 765.
  2. A. H. Osman, A. A. Alzahrani, J. New Approach for Automated Epileptic Disease Diagnosis Using an Integrated Self-Organization Map and Radial Basis Function Neural Network Algorithm. IEEE Access 2019, vol. 7, pp. 4741–4747, doi: 10.1109/ACCESS.2018.2886608.

The revised content is the highlighted part of the latest version of the manuscript named " sensors-898789-r02.pdf".

Kind regards,

Yan Liu

Reviewer 3 Report

Thank you for interesting paper. Your topic - prediction of ischemic stroke probability by neural networks – is very promising and challenging.

I have few suggestions from clinical viewpoint regarding the risk of stroke that might to improve the value of your research.

  1. I wonder why you selected EMG (electromyography) as one of sources for streaming data. EMG is the classical neurological investigation of various human muscles. This study has long been used to study a variety of human skeletal muscles in myopathies, neuropathies, and other peripheral diseases of the nervous system. However, I am not aware about the use of EMG for predictiction of stroke risk. You also don’t mention it in the background or discussions. Could you provide any references about relationship between EMG findings and stroke? In my opinion, echocardiography data would be much more reasonable.
  2. ECG and especially EMG studies provide a lot of numeric data as study findings (for example, EMG - latencies, signal amplitude, jitters, etc., and these findings are dependent on which muscles are being examined. In Methods you should more thoroughly describe the methodology of EMG study (examined muscles, measurements, etc.) if you use this study for your model at all (see previous comment #1).
  3. Minor grammar revisions are needed (for example, line 24 "effiectively", line 25 "Acutally", etc.).

Author Response

Dear reviewer,

Many thanks for these valuable suggestions. According to your suggestions, we revised the manuscript. The changes we made are as follows:

#1. Based on the references that include the introduction of the relationship between stroke and EMG, we propose that stroke will have a certain degree of impact on EMG in the ' Introduction' section on page 2, on lines 67-71. Therefore, EMG can also be used as a predictive factor for stroke prediction. Modifications in the 'Introduction' section as follows: Di [14] recruited 11 post-stroke patients and 20 healthy control subjects and performed an elbow sinusoidal trajectory tracking experiment. The experimental results showed that the EMG's fApEn (fuzzy approximate entropy) values of the experimental group and the control group were significantly different, so stroke can induce neurological changes in paretic muscles.

  1. D. Ao, R. Sun, K. Y. Tong, and R. Song, J. Characterization of Stroke- and Aging-Related Changes in the303 Complexity of EMG Signals During Tracking Tasks.J. Ann. Biomed. Eng. 2015 ,vol. 43, no. 4, pp. 990–1002,304 doi: 10.1007/s10439-014-1150-1

#2. We have described the methodology of EMG study in this paper, as shown in Table 1 on page 3.

#3. We also corrected the grammatical errors in the manuscript.

The revised content is the highlighted part of the latest version of the manuscript named " sensors-898789-r03.pdf".

Kind regards,

Yan Liu

Round 2

Reviewer 1 Report

The authors have sufficiently addressed most of my concerns.

There are several grammatical errors throughout the manuscript and I suggest they be reviewed prior to re-re-submission.

Reviewer 2 Report

Minor spellchecking is required. Otherwise the paper is ok.